# A Theory for Estuarine Delta Formation with Finite Beach Length under Sediment Supplied from the River

Dinh Van Duy [1],*, Hitoshi Tanaka [2], Magnus Larson [3] and Nguyen Trung Viet [4]

1 Division of Hydraulics Engineering, Can Tho University, Can Tho 94000, Vietnam
2 Institute of Liberal Arts and Sciences, Tohoku University, Sendai 980-8576, Japan; hitoshi.tanaka.b7@tohoku.ac.jp
3 Division of Water Resources Engineering, Lund University, 221 00 Lund, Sweden; magnus.larson@tvrl.lth.se
4 Department of Civil Engineering, Thuyloi University, Ha Noi 11515, Vietnam; nguyentrungviet@tlu.edu.vn
* Correspondence: dvduy@ctu.edu.vn

**Abstract:** Analytical solutions for a one-line model for shoreline changes are employed to investigate the formation of two wave-dominated river delta coastlines, along with a small-scale laboratory experiment. Since the present analytical solution can be applied only to a river delta with infinite shorelines, a new analytical solution was developed to consider the effects of lateral boundaries to the evolution of delta coastlines. It was determined that two demarcations represented by the dimensionless times $t^*$ can be used to judge whether the lateral boundaries have affected the coastline evolution or not. After the successful application of a new analytical solution to the experimental data, the new analytical solution was applied to predict the formation and deformation of the shorelines of the Ombrone River and Funatsu River deltas. Results obtained from the analysis showed that the new analytical solution can be used to describe well the formation and deformation of finite river-delta shorelines. Based on the two demarcations as represented by the dimensionless time $t^*$, the shorelines of the Ombrone River and Funatsu River deltas are classified as finite shorelines.

**Keywords:** analytical solution; finite shorelines; river sediment; river delta; wave-dominated; formation; deformation

## 1. Introduction

River deltas are classified as a lowland area formed by the prolonged accumulation of river-borne sediment. This low-topography area may provide a wide range of ecosystem services such as coastal defense, drinking water supply, and tourism, plus industry and transport, which lead to major urbanization activities [1]. Due to their potentially rich economic and ecological functions, river deltas are of considerable interest to many researchers, as evidenced by the tremendous number of publications around the world in the last few decades [1–11].

Human intervention in a coastal area and a river basin often results in the coastal erosion of the river-delta shorelines, especially near the river mouth. For example, damming rivers for hydropower has been reported as one of the most common problems that results in the severe erosion of downstream beaches. Recent coastal erosion of river-delta shorelines has been reported on the global scale, including those of the Nile [2,12], Mississippi [13,14], Yangtze [14], Tenryu [15,16], and Thu Bon [17,18] rivers.

As coastal erosion is happening on a global scale and threatening the hundreds of millions of people living in deltas today [1], it is necessary to take into consideration the morphological changes of river-delta coastlines. Unfortunately, inadequate monitoring of many rivers around the world has limited the ability to identify the impacts of coastal erosion at the river mouths [16]. It is, therefore, necessary to employ a method that can be used as a starting point for any coastal project, to give a comprehensive discussion about the fundamental beach behavior under simplified initial and boundary conditions. For

that reason, an analytical approach is preferable since it requires less measured data for the calculations.

The processes of the formation of and reduction in river deltas have been studied through the theory of the one-line model, with analytical solutions of diffusion types. Penarld-Considere (1956) [19] was the first to introduce the theory of the one-line model, and it has been proven to be adequate in practical applications. Since then, several improvements have been made for the analytical solution of shoreline change under different conditions [20–27]. Fundamentally, the assumptions that comprise the theory are as follows: (1) the beach profile moves parallel to itself, (2) longshore sediment transport take place uniformly over the beach profile down to the limited depth of littoral drift, (3) details of nearshore circulation are neglected, and (4) the longshore-sediment transport rate is proportional to the breaking-wave properties.

Although there are several analytical solutions for studying the formation and deformation of river deltas, there is one common limitation of these analytical solutions, which is the assumption of infinite beach length.

Therefore, the objective of this study is to propose a new approach on how to incorporate the effect of lateral boundaries into the analytical solution of [3]. Among the river deltas in the world, two river deltas with different scales including the Ombrone River delta in Italy and the Funatsu River delta in Japan, which are selected as the subject of this study to give a comprehensive perspective about the formation of river-delta shorelines. The reasons for selecting these two river deltas are (i) they are wave-dominated river deltas [28] that are suitable for the application of analytical model, (ii) the availability of data for the analysis, and (iii) they show different spatial scales of shoreline length, which is a very important parameter in this study. Based on the analysis of field data as well as laboratory experimental data, classification has been made for the river-delta shorelines into two categories, which are the finite and infinite shorelines, based on their temporal and spatial scales.

## 2. Theory for Delta Formation

### 2.1. One-Line Model

In a one-line model, the beach profile is assumed to move uniformly, as illustrated in Figure 1. In other words, there are no changes in the shape of the beach during its landward or seaward movements in response to erosion and accretion, respectively. An important geometrical aspect in the one-line theory is the profile height, which has a landward limit at the top of the beach berm ($D_B$), and the seaward limit, where there is no significant change of the depth, which is often referred to as the so-called depth of closure ($D_C$).

Based on the aforementioned ideas, any point on the beach profile can be used to represent this profile and, as a result, one contour line can be used to sufficiently represent the beach-plan shape for the computation of shoreline change or beach-volume change. This provides a simple relationship between the shoreline change and the spatial variation of longshore-sediment-transport rate as:

$$\frac{\partial y}{\partial t} + \frac{1}{D}\frac{\partial Q}{\partial x} = 0 \tag{1}$$

where $x$ is the alongshore coordinate, $y$ is the shoreline position, $t$ is the time, $D = D_B + D_C$ ($D_B$: berm height, $D_C$: depth of closure), and $Q$ is the longshore-sediment-transport rate.

According to [3], Equation can be simplified to a linear differential equation if the amplitude of the longshore-sand-transport rate and the incident-breaking-wave angle are constant and small:

$$\frac{\partial y}{\partial t} = \varepsilon \frac{\partial^2 y}{\partial x^2} \tag{2}$$

where $\varepsilon$ is the diffusion coefficient expressed as:

$$\varepsilon = \frac{K(H^2 C_g)_b}{8}\left(\frac{\rho}{\rho_s - \rho}\right)\left(\frac{1}{1-n}\right)\left(\frac{1}{D_B + D_C}\right) \tag{3}$$

where $K$ is the dimensionless coefficient for sediment transport rate formula, $H$ is the wave height, $C_g$ is the group velocity, subscript "$b$" denotes the quantity at the wave-breaking point, $\rho$ and $\rho_s$ are, respectively, the density of sea water and sand, and $n$ is the sediment porosity ($n = 0.4$). According to Equation (3), the diffusion coefficient is highly dependent on the breaking-wave height ($H_b$) and the dimensionless empirical coefficient for longshore-sediment transport ($K$).

The wave height and velocity at the breaking point can be calculated as [29]:

$$H_b = kg^{1/5}\left(TH_\infty^2\right)^{2/5} \tag{4}$$

$$C_{gb} = \sqrt{\frac{gH_b}{\kappa}} \tag{5}$$

where $K$ is the dimensionless coefficient, $K = 0.39$, and $T$ and $H_\infty$ are, respectively, the wave period and deep-water-wave height, $\kappa = 0.78$.

It is noted that Equation (2) is identical to the one-dimensional equation of heat conduction or the diffusion equation. Therefore, by applying the proper analogies between the initial and boundary conditions for shoreline evolution and the processes of heat conduction, many analytical solutions for shoreline change can be obtained.

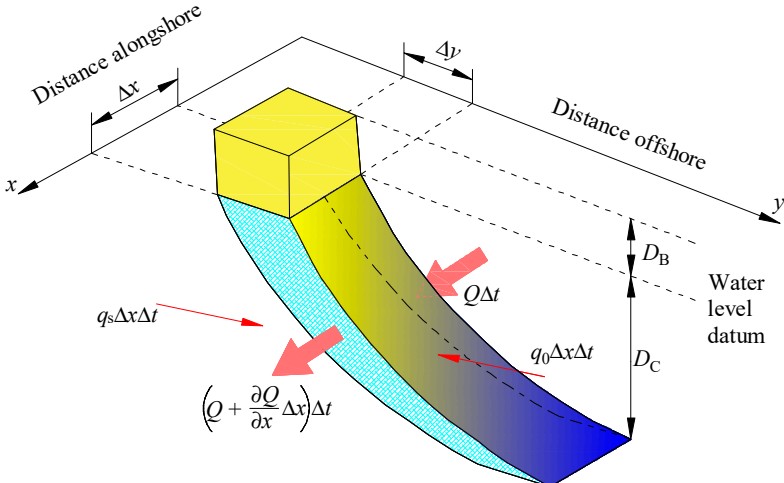

**Figure 1.** Definition sketch for shoreline-change calculation [30].

### 2.2. A Theory for Delta Formation with Infinite Sandy-Beach Length

Ref. [3] proposed an analytical solution that can be used to represent the shoreline evolution in the vicinity of a river discharging sand on an infinite-length beach, as defined in Figure 2a. The formation process of a river delta located on a sandy beach with infinite length can be represented as:

$$y = f_1(x,t) = \frac{q_0}{D}\sqrt{\frac{t}{\pi\varepsilon}}e^{-x^2/(4\varepsilon t)} - \frac{q_0}{D}\frac{|x|}{2\varepsilon}erfc\left(\frac{|x|}{2\sqrt{\varepsilon t}}\right) \tag{6}$$

where $q_0$ is the sediment-supply rate from the river, and *erfc* is the complementary error function. From Equation (6), the maximum shoreline position $y_0$ at the river mouth can be expressed as:

$$y_0 = \frac{q_0}{D}\sqrt{\frac{t}{\pi\varepsilon}} \tag{7}$$

Thus, Equation (6) can be expressed in the dimensionless form as:

$$\eta = \exp\left(-\xi^2\right) - \sqrt{\pi}\xi erfc(\xi) \tag{8}$$

where

$$\eta = \frac{y}{y_0} \tag{9}$$

and

$$\xi = \frac{|x|}{2\sqrt{\varepsilon t}} \tag{10}$$

Using Taylor series expansion, the polynomials of degree 1 and 2 of Equation (8) are expressed as:

$$\eta = 1 - \sqrt{\pi}\xi \tag{11}$$

$$\eta = 1 - \sqrt{\pi}\xi + \xi^2 \tag{12}$$

It is interesting to note that, although Equation (6) is expressed in term of various variables, Equation (8) gives a highly simple relationship between $\xi$ and $\eta$. Figure 3 shows the shape of the delta given by Equations (8), (11), and (12). The diagram is plotted only for $\xi \geq 0$, since the solutions are symmetric with respect to the $\eta$-axis. It is seen that Equation (12) gives a reasonably good result for $\xi < 0.5$.

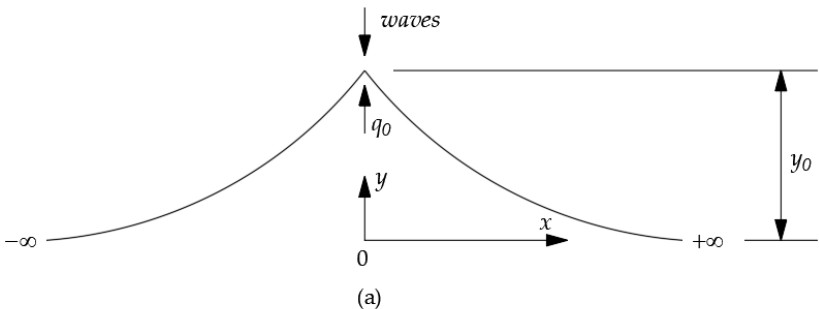

(a)

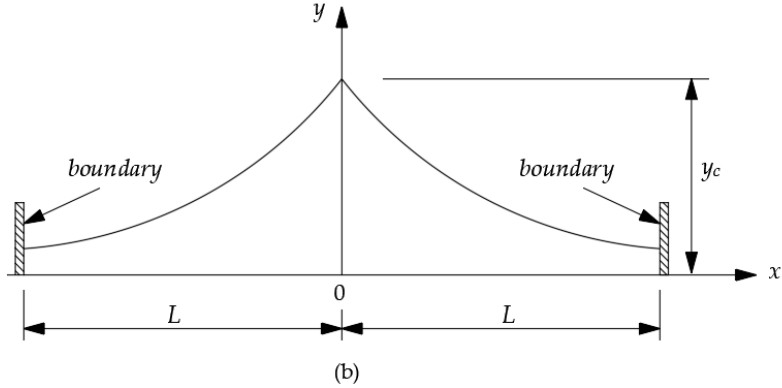

(b)

**Figure 2.** Schematic diagram for delta-shoreline formation owing to sediment supply from the river as a point source; (**a**) infinite-length beach, (**b**) finite-length beach.

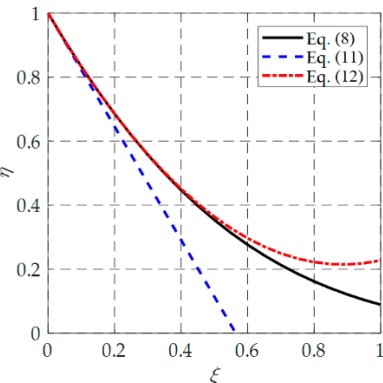

**Figure 3.** Schematic diagram for delta-shoreline formation owing to sediment supply from the river as a point source.

### 2.3. A Theory for Delta Formation with Finite-Sandy-Beach Length

It should be noted that for the analytical solution of [3], Equation (6) is applicable for shorelines of infinite lengths. In reality, however, the shorelines are always finite due to the existence of headlands or coastal structures located at distances from the river mouth. Therefore, it would be useful to derive another analytical solution that can be used to study the evolution of a river delta with finite shorelines, as schematized in Figure 2b.

Although there has not been any analytical solution for the development of a river-delta coastline with a finite length, there is a comparative study about the heat conduction in a medium between two insulated boundaries [31,32]. With reference to the books [31,32], a new equation for shoreline evolution under the effect of no-transport boundaries can be obtained as:

$$y = f_2(x,t) = \frac{q_0}{2\varepsilon DL}\left[\frac{x^2}{2} - L|x| + \frac{L^2}{3} + \varepsilon t - \frac{2L^2}{\pi^2}\sum_{n=1}^{\infty}\frac{1}{n^2}e^{-n^2\pi^2\frac{\varepsilon t}{L^2}}\cos\left(\frac{n\pi x}{L}\right)\right] \quad (13)$$

where $L$ is the length of the shoreline, as defined in Figure 2b. Similar to Equation (8), Equation (13) can be expressed in term of dimensionless quantities as follows.

$$y^* = \frac{x^{*2}}{2} - |x^*| + \frac{1}{3} + t^* - \frac{2}{\pi^2}\sum_{n=1}^{\infty}\frac{e^{-n^2\pi^2 t^*}}{n^2}\cos(n\pi x^*) \quad (14)$$

where

$$y^* = y\frac{2\varepsilon D}{q_0 L} \quad (15)$$

$$x^* = \frac{x}{L} \quad (16)$$

$$t^* = \frac{\varepsilon t}{L^2} \quad (17)$$

In order to observe the differences between the new analytical solution and the analytical solution provided by [3], Equation (6) is transformed into the dimensionless form using Equations (15)–(17), which are different definitions as compared with Equations (9) and (10).

$$y^* = 2\sqrt{\frac{t^*}{\pi}}e^{-\left(\frac{x^{*2}}{4t^*}\right)} - |x^*|erfc\left(\frac{|x^*|}{2\sqrt{t^*}}\right) \quad (18)$$

Figure 4 shows the comparison between the new analytical solution and Equation (18). As can be seen in Figure 4, when $t^*$ is smaller than 0.1, Equations (14) and (18) are in perfect agreement. From around $t^* = 0.2$, a difference starts to appear at the right end boundary. However, there is no difference at the river mouth. Thereafter, the difference between the

two solutions has expanded, and the influence of the lateral boundary is clear. After $t^* = 0.4$, the shoreline of parabolic shape is moving forward in the offshore direction.

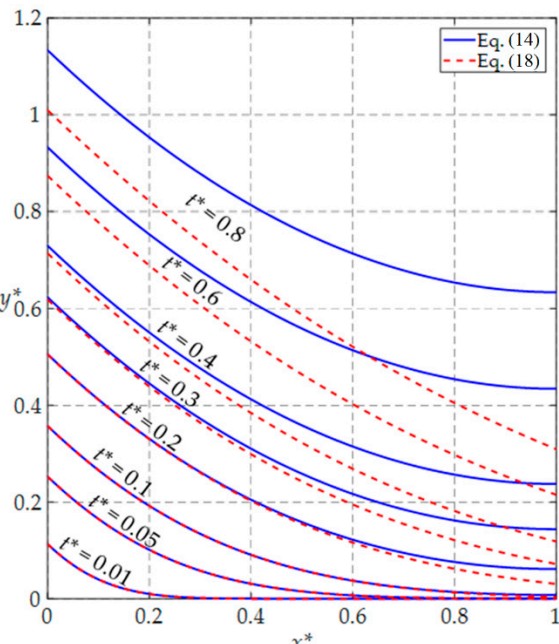

**Figure 4.** Comparison of shoreline positions calculated by two solutions.

The behaviors of the two solutions in Figure 4 can be explained by examining two asymptotes at $t^* \to 0$ and $t^* \to \infty$. When $t^*$ is small, sediment from the river has not reached the lateral boundary, and there is no effect of the boundary. Therefore, the behavior of the shoreline from Equations (14) and (18) must be the same. However, at large value of $t^*$, the exponential term in Equation (14) vanishes, and the shoreline with a parabolic shape will move seaward at the constant propagation speed, $t^*$, as:

$$y^* = \frac{x^{*2}}{2} - |x^*| + \frac{1}{3} + t^* \tag{19}$$

Figure 5a shows the shoreline evolution at the river mouth ($x^* = 0$), where the shoreline position can be expressed as:

$$y_0^* = t^* + \frac{1}{3} - \frac{2}{\pi^2} \sum_{n=1}^{\infty} \frac{e^{-n^2 \pi^2 t^*}}{n^2} \tag{20}$$

As can be seen in Figure 5a, when $t^*$ is small, the new analytical solution and Equation (18) are in perfect agreement, which indicates that there is no influence by the lateral boundary on the shoreline evolution at the river mouth. Hence, the shoreline evolution at the river mouth when $t^*$ approaches 0 can be represented as:

$$y_0^* = 2\sqrt{\frac{t^*}{\pi}} \tag{21}$$

When $t^*$ is larger, the difference between Equations (14) and (18) can be observed in Figure 5a. This indicates the influence of the lateral boundary to the shoreline evolution at the river mouth after a sufficient time. At this stage, the shoreline evolution at the river mouth is given by Equation (14) can be expressed as $t^* \to \infty$:

$$y_0^* = t^* + \frac{1}{3} \tag{22}$$

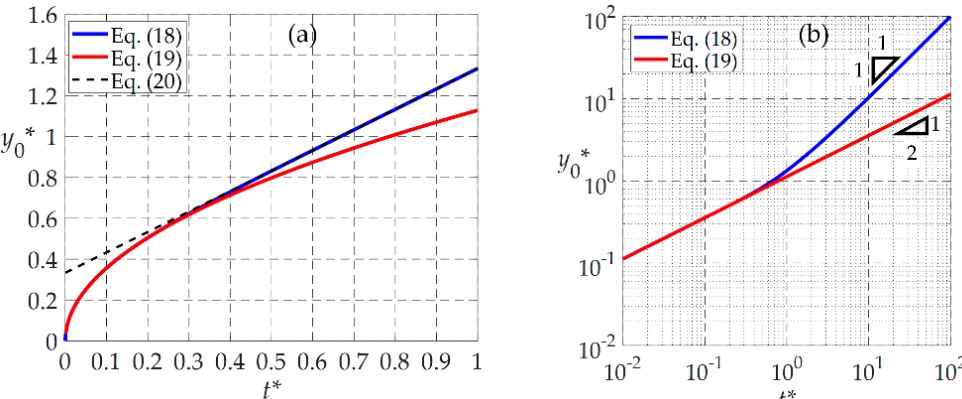

**Figure 5.** Shoreline evolution at the river mouth with (**a**) normal plot and (**b**) log-log plot to show the difference of power to the dimensionless time.

Figure 5b is plotted with the same relationship using the logarithmic axes to show the difference of power to the dimensionless time in Equations (21) and (22). In Figure 5, it should be noted that the transit time between Equations (21) and (22) exists around $t^* = 0.3$.

Meanwhile, Figure 6 shows the shoreline evolution at the boundary, which can be expressed as $x^* = 1$ in Equation (14).

$$y_1^* = t^* - \frac{1}{6} - \frac{2}{\pi^2} \sum_{n=1}^{\infty} (-1)^n \frac{e^{-n^2 \pi^2 t^*}}{n^2} \tag{23}$$

By taking the limits as $t^* \to \infty$, the shoreline evolution at the lateral boundary for large $t^*$ is obtained from Equation (23) as:

$$y_1^* = t^* - \frac{1}{6} \tag{24}$$

If we use the solution for infinite-beach length, Equation (18), the shoreline position at $x^* = 1$ is obtained as:

$$y_1^* = 2\sqrt{\frac{t^*}{\pi}} e^{-\left(\frac{1}{4t^*}\right)} - erfc\left(\frac{1}{2\sqrt{t^*}}\right) \tag{25}$$

As can be seen in Figure 6, the transit time between Equations (23) and (25) exists near $t^* = 0.1$.

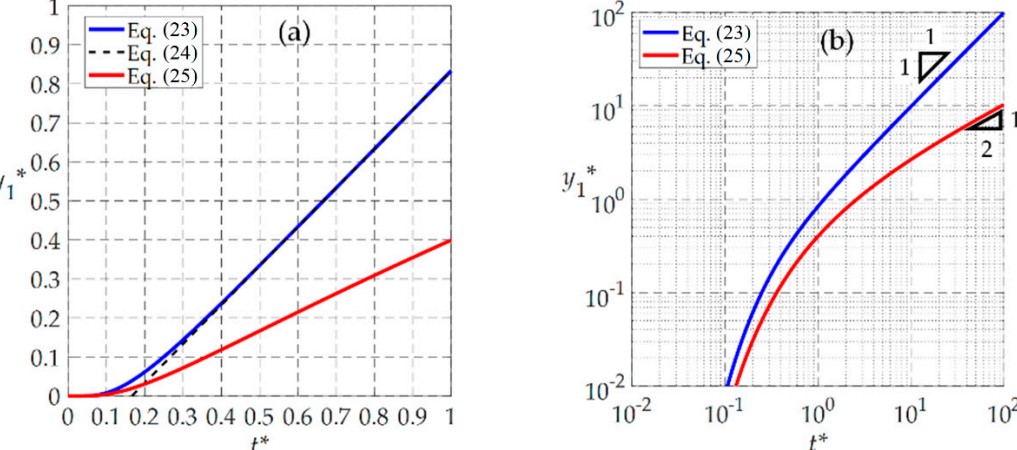

**Figure 6.** Shoreline evolution at the boundary with (**a**) normal plot and (**b**) log-log plot to show the difference of power to the dimensionless time.

Next, the corresponding distribution of longshore sediment transport rate will be discussed. The longshore-sediment-transport rate can be obtained from the gradient of the shoreline.

$$Q = -\varepsilon D \frac{\partial y}{\partial x} \tag{26}$$

Substituting Equation (13) into Equation (26), the dimensionless longshore sediment, $Q^*$, is obtained as,

$$Q^* = \text{sgn}(x^*) - x^* - \frac{2}{\pi^2} \sum_{n=1}^{\infty} \frac{e^{-n^2\pi^2 t^*}}{n} \sin(n\pi x^*) \tag{27}$$

where sgn is the sign function, and $Q^*$ is defined by the following equation:

$$Q^* = \frac{Q}{q_0/2} \tag{28}$$

If $t^* \to 0$, one of the asymptotic solutions is obtained from Equation (18)

$$Q^* = erfc\left(\frac{|x^*|}{2\sqrt{t^*}}\right) \tag{29}$$

Meanwhile, if $t^* \to \infty$, the corresponding solution is obtained from Equation (27), indicating the linear distribution of the sediment movement from the river mouth to the beach end.

$$Q^* = \text{sgn}(x^*) - x^* \tag{30}$$

The sediment-transport rate from Equation (27) is illustrated in Figure 7, in which, if $t^* < 0.05$, Equation (27) exactly coincides with Equation (29). It means the sediment movement does not reach the rigid boundary, whereas with the increase in $t^*$, the profile of the sediment movement approaches the linear distribution given by Equation (30).

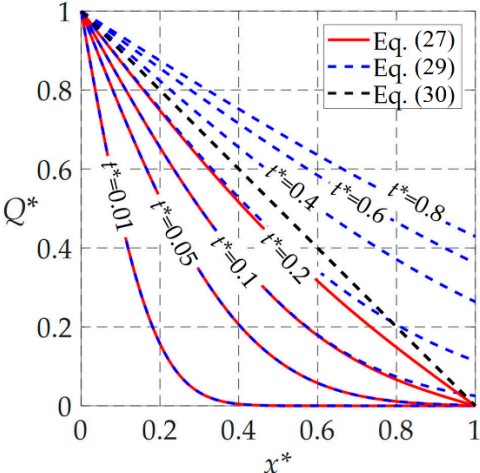

**Figure 7.** Dimensionless longshore-sediment-transport rate $Q^*$.

*2.4. A Theory for Delta Deformation with Infinite- and Finite-Sandy-Beach Length*

Using a principle linear superposition [17,33], the deformation of a delta with infinite and finite shorelines can be expressed as:

For the infinite shorelines:

$$y = f_1(x, t) + R f_1(x, t - t_1) \tag{31}$$

For the finite shorelines:

$$y = f_2(x, t) + R f_2(x, t - t_1) \tag{32}$$

Where $f_1$ and $f_2$ are defined in Equations (6) and (13), respectively, $R$ is the reduction rate of the sediment supply from the river, and $t_1$ is the time when the erosion started in the delta.

## 3. Application of the Theory to a Laboratory Experiment

In the previous section, the new analytical solution for the formation of a river delta with finite shorelines has been introduced. In order to validate the new analytical solution, experimental results from [2] are utilized. This experiment was conducted to study the formation process of a river delta in a wave basin that is 8 m wide, as depicted in Figure 8. Two waveguide walls are located at about 3.5 m on both sides of the wave basin. These two waveguide walls can be considered as the lateral boundaries that lead to the formation of finite shorelines. A sediment-feeder machine was used to automatically discharge lightweight aggregate as a point source to the wave basin, at a constant rate of 7.06 cm$^3$/s in 80 min. Waves were set to approach normally to the shoreline. The wave height and period are 2.0 cm and 0.8 s, respectively. The water depth was set constantly at 30 cm. The shoreline positions were recorded at 10 min intervals along the delta at 13 stations (50 cm interval distance) in Figure 8. Since lightweight aggregate was used as the sand in the experiment of [2], the value of the sediment density ($\rho_s$) can be referenced to the study of [34].

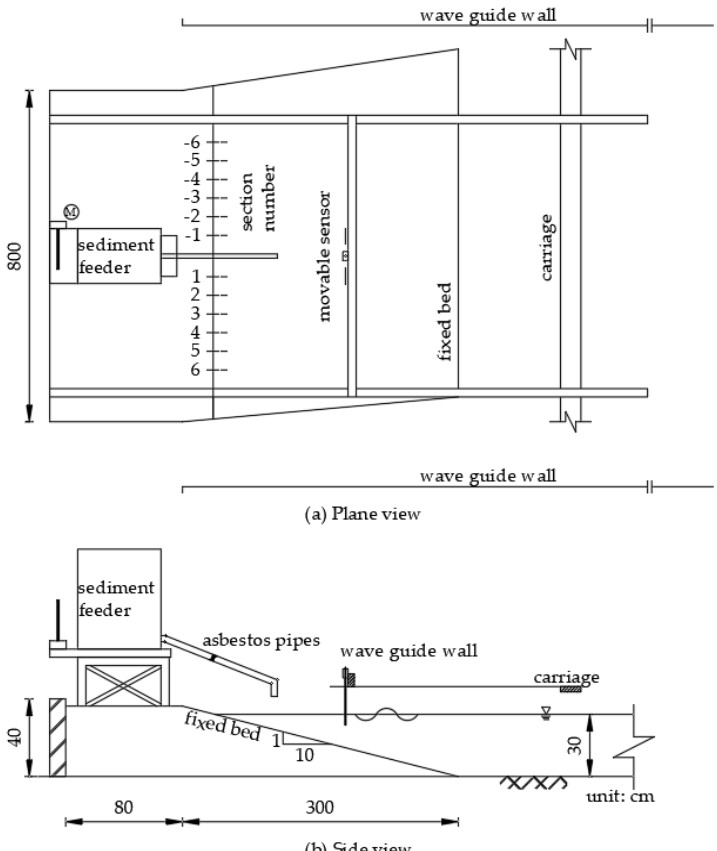

**Figure 8.** Schematic diagram of experiment for formation of river deltas [2]. (**a**) Plane view; (**b**) Side view.

In order to simulate the formation process of the delta using Equations (6) and (13), the diffusion coefficient $\varepsilon$ must be determined. The value of $\varepsilon$ is calibrated by making a comparison between the calculated and the measured shoreline evolutions at the river mouth, as shown in Figure 9, in which the calculated shoreline evolutions were accomplished by using

Equation (7) with different proposed values of $\varepsilon$. As can be seen in Figure 9, the shoreline evolution calculated by $\varepsilon = 15$ cm$^2$/s shows good agreement with the experimental data.

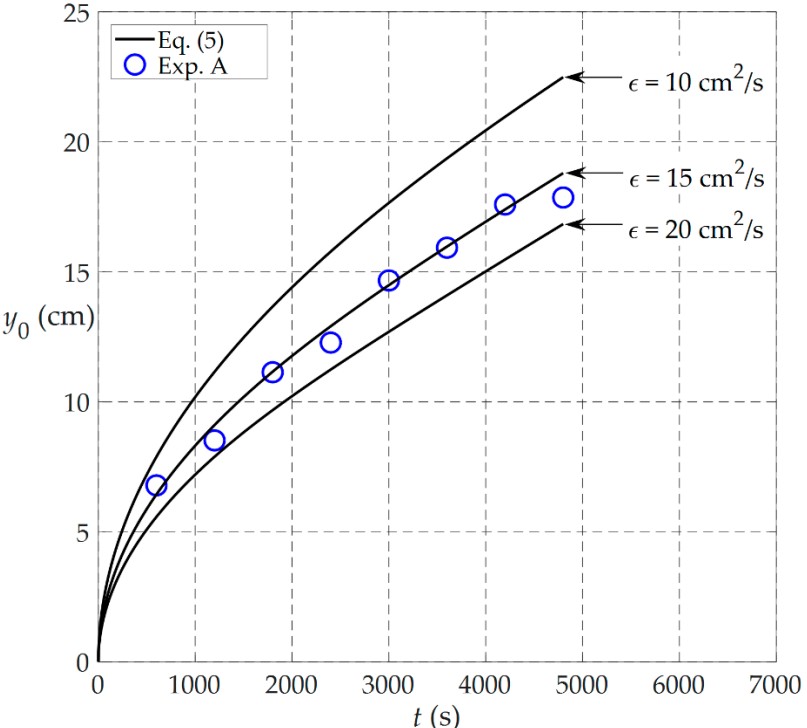

**Figure 9.** Calibration of $\varepsilon$ for the experimental data from the study of [2].

Since the wave data were known in the experiment, it would be useful to make discussion between $\varepsilon$ and $K$ as shown in Equation (5). Using Equation (5) and the ralated values (as shown above), $\varepsilon$ can be determined as 14 cm$^2$/s. This value shows good agreement with the value of $\varepsilon = 15$ cm$^2$/s obtained from the fitting method shown in Figure 9. This shows the reliable of the fitting method since the value of $K = 0.39$ [35] was used to obtain $\varepsilon = 14$ cm$^2$/s.

The recorded beach profiles in the experiment were also used to obtain the value of $D_C = 3.9$ cm. As can be seen in Figure 10, the new solution achieves better agreement with the experimental results, especially near the boundary, $x = 300$ cm.

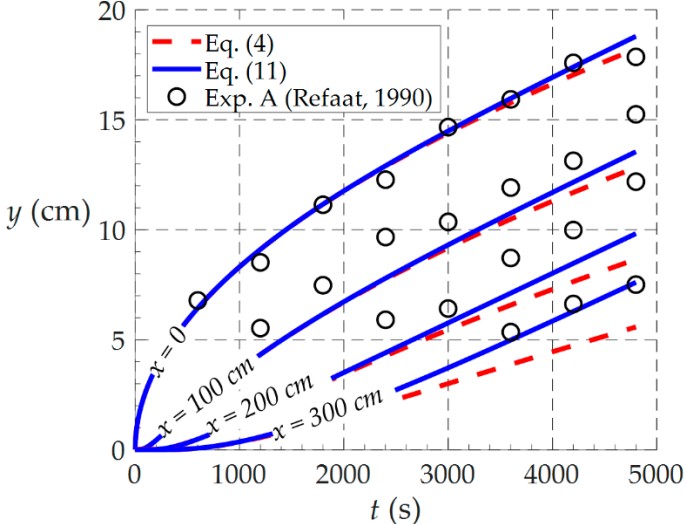

**Figure 10.** Comparison between theoretical and experimental data.

## 4. Application of the Theory to Field Data

### *4.1. Study Area*

In this section, two study areas will be introduced, including the Ombrone River delta in Italy and the Funatsu River delta in Lake Inawashiro, Japan, as summarized in Table 1. Of these, the Ombrone River delta is an open-coast river delta while the Funatsu River delta is formed in a lake. Therefore, investigation of these two river deltas can sufficiently give a general picture about the formation of the wave-dominated river deltas in the world.

**Table 1.** Shorelines data used in this study.

| No. | River | Length of the Shorelines (km) | |
| | | Left | Right |
| --- | --- | --- | --- |
| 1 | Ombrone (Italy) | 6.0 | 16 |
| 2 | Funatsu (Japan) | 0.5 | 4.5 |

### 4.1.1. Ombrone River Delta

Figure 11 shows a map of the Ombrone River basin located in central Italy. The Ombrone River has a steep basin, with the average relief of approximately 250 m and a small catchment area of about 3500 km$^2$. The annual precipitation in this basin is 800 mm. In its steep basin, the Ombrone River flows over a distance of 130 km before discharging to the Tyrrhenian Sea. From the study of [36], the total sediment input from the Ombrone River is $1.35 \times 10^6$ m$^3$/y, in which only 30% of this total volume contributes to the morphological changes of the delta's lobes, where the beaches consist of mostly gravel and sand [36]. Hence, the value of $q_0 = 405,000$ is used as the initial condition for the calibration, as presented in Table 2.

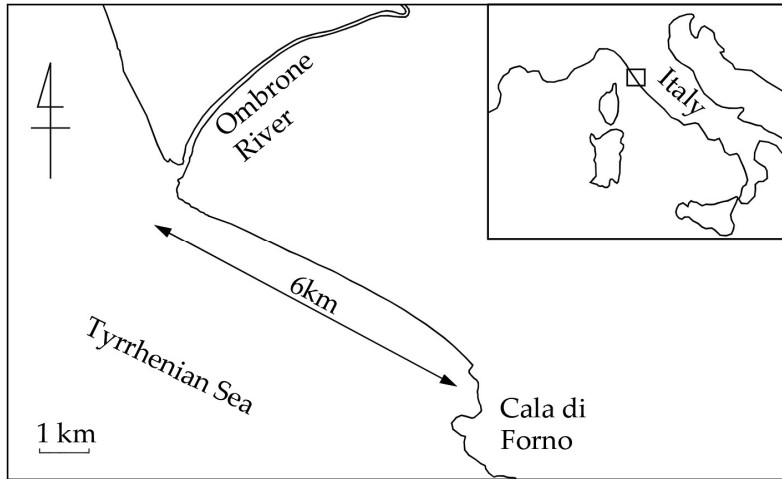

**Figure 11.** Ombrone River delta in Italy.

The Ombrone River delta is one of the major Tyrrhenian deltas and the most natural area in the Tuscany region that has not been much disturbed by humans, regarding both the delta's coastal area and river basin [37,38].

The Ombrone River delta has a coastline of about 30 km, running from Castiglione Della Pescaia in the north to Cala di Forno in the south. In the present study, the morphology change from the mouth to Cala di Forno will be analyzed.

### 4.1.2. Funatsu River Delta

As can be seen in Figure 12, the Funatsu River delta is formed by the sediment discharged from Funatsu River to Lake Inawashiro. Lake Inawashiro is located in Fukushima Prefecture. According to [39], the catchment area of Funatsu River is 61.5 km$^2$. It is clear

that the Funatsu River delta's coastline has a smaller scale than the Ombrone River delta's coastline, as seen in Table 1. Therefore, study of the formation processes of the Funatsu delta shorelines is useful because the lakeshores can be considered as an intermediate between coastal zones and laboratory experiments [40].

As this delta is formed in a lake, its spatial scale is very small, with the shoreline on the left of the Funatsu River measuring approximately 490 m, which is bounded by a port structure, as seen in Figure 12.

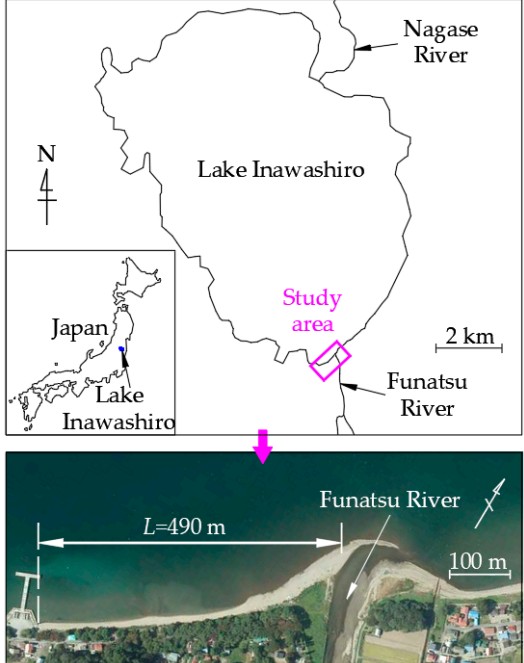

**Figure 12.** Funatsu River delta in Lake Inawashiro, Japan.

*4.2. Application of the Theory to the Ombrone River Delta*

Figure 13 shows the map and progradation of the Ombrone River delta with morphological features and shoreline changes [38].

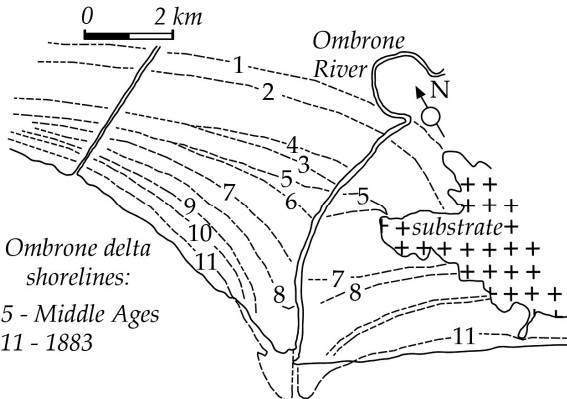

**Figure 13.** Progradation of the Ombrone River delta and shoreline changes (1–11) [38].

The formation process of the Ombrone River delta is calculated using the aforementioned theory for finite-beach length, in which Equation (13) is used to calculate the shoreline positions. The parameters required for the application of Equation (13) are summarized in Table 2.

**Table 2.** Initial values for the calibration condition.

| Parameters | Values |
| --- | --- |
| Diffusion Coefficient, $\varepsilon$ (m$^2$/day) | Unknown |
| Sediment supply from the river, $q_0$ (m$^3$/y) | 405,000 [36] |
| Formation time, $t_0$ (yr) | 1000 [38] |
| Depth of closure, $D_C$ (m) | 8 [41] |
| Berm height, $D_B$ (m) | 1 [41] |
| Length of the shoreline, $L$ (m) | 6000 |

In Table 2, the value of $\varepsilon$ will be calculated using the fitting method. In the fitting method, the calculated shoreline in 1883 is compared with the measured shoreline in 1883, because the shoreline position in 1883 is considered as the final one in the formation process of the Ombrone River delta [42–44].

In order to reduce the iteration of the calculation, the values of sediment supply from the river ($q_0$) and the formation time ($t_0$) are initially set at 405,000 m$^3$/y [36] and 1000 years [38], respectively.

Table 3 shows the calibrated values for the Ombrone River delta. The calculated and the measured shorelines in 1883 are shown in Figure 14. For each calculation, the calculated shoreline in 1883 is compared with the measured shoreline in 1883.

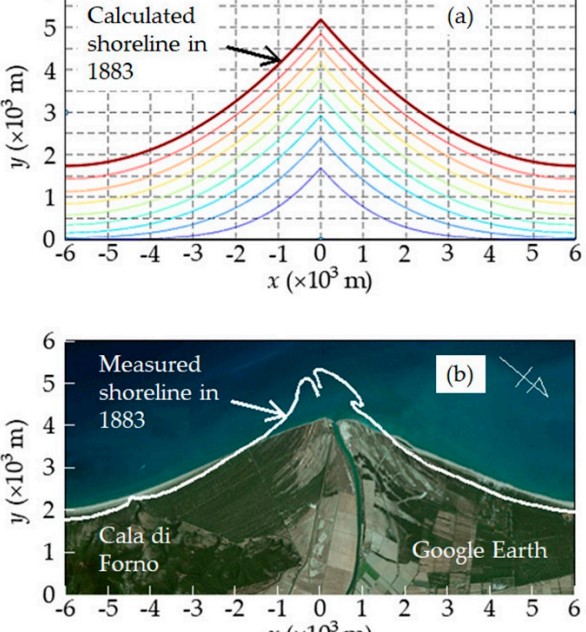

**Figure 14.** Calculated and measured shorelines in 1883 of the Ombrone River delta. The time interval for calculated shorelines in (**a**,**b**) is 100 years.

The RMSE is calculated, and the calculation will stop when the minimum RMSE is obtained. Figure 15 shows the best fit between the calculated shoreline using Equation (14), and the measured shoreline in 1883 corresponds to the minimum RMSE = 130 m. It should be noted that the comparison is only for the left shoreline.

The calibrated values determined from the fitting process are presented in Table 3. Those values are validated by applying them to the deformation of the Ombrone River delta, with the reduction rate of sediment supply from the river $R$ = 0.75 [43] and the starting point of erosion $t_1$ = 1883 [44].

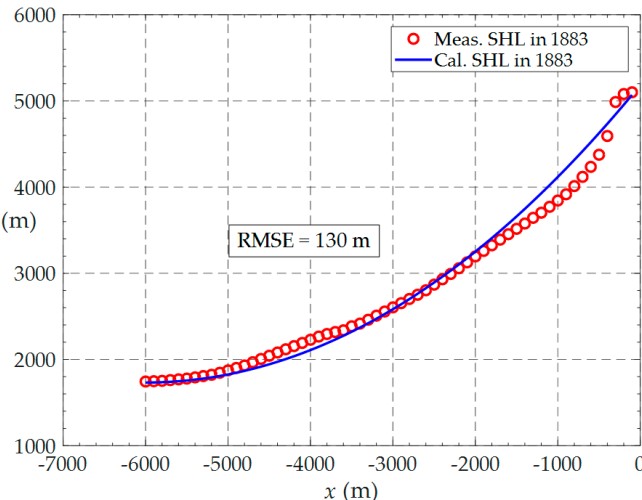

**Figure 15.** Best fit between calculated and measured shorelines in 1883 of the Ombrone River delta shorelines.

**Table 3.** Calibrated values determined from the fitting process for the Ombrone River delta.

| Parameters | Values |
|---|---|
| Diffusion Coefficient (Calibrated), $\varepsilon$ (m²/day) | 45 |
| Sediment supply from the river, $q_0$ (m³/y) | 345,000 |
| Formation time, $t_0$ (yr) | 900 |
| Depth of closure, $D_C$ (m) | 8 [41] |
| Berm height, $D_B$ (m) | 1 [41] |
| Length of the shoreline, $L$ (m) | 6000 |
| Reduction rate, $R$ | 0.75 [43] |
| Time when erosion happened, $t_1$ | 1883 [44] |

The validation is shown in Figure 16, in which Figure 16a shows the shoreline positions at the river mouth and Figure 16b shows the shoreline position at the boundary. It can be seen that the shorelines calculated by the new analytical solution show better agreement with the measured data.

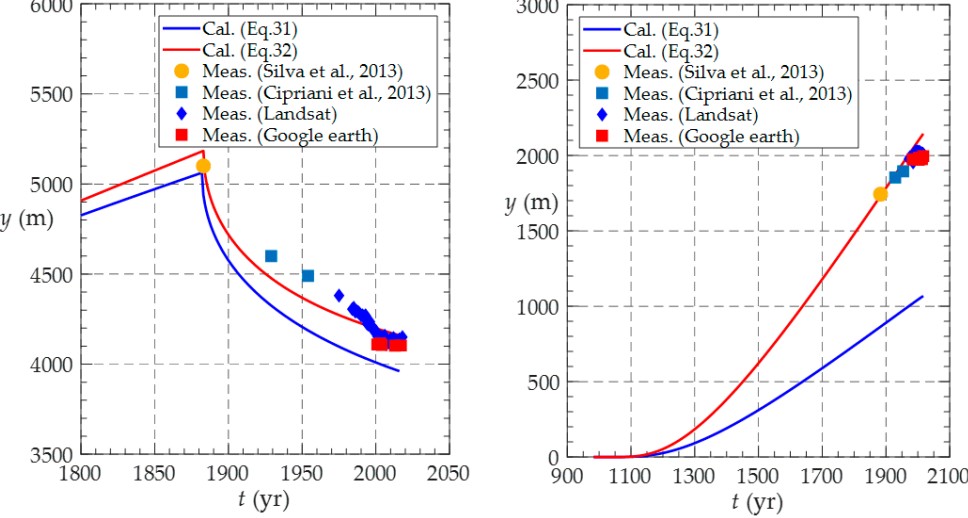

**Figure 16.** Validation of the calibrated parameters for the Ombrone River delta at (**left**) the river mouth and (**right**) at the boundary.

### 4.3. Application of the Theory to the Funatsu River Delta

As can be seen in Equation (17), the dimensionless time *t** that determines the effect of the boundary is the function of the shoreline length, *L*, and the diffusion coefficient, *ε*. Therefore, it would be useful to examine the formation of a river-delta lakeshore that has smaller scales in both the external forces and the geometry.

In Figure 12, the left shoreline of the Funatsu River delta can be considered as a finite extent between the river mouth and the pier located at 490 m from the river mouth. Therefore, the formation of this shoreline will be examined using the new analytical solution for the formation of finite-delta shorelines, Equation (13).

Using the same procedure as in the calculation for the Ombrone River delta, the initial and calibrated data for the Funatsu River delta can be obtained, as shown in Table 4, in which the total of the depth of closure ($D_C$) and the berm ($D_B$) height is determined from the relationship between the cross-sectional change ($\Delta A$) and the shoreline change ($\Delta y$) [45].

**Table 4.** Initial and calibrated data determined by applying the fitting process to the Funatsu River delta.

| Parameters | Initial Data | Calibrated Data |
|---|---|---|
| Diffusion coefficient, $\varepsilon$ (m$^2$/day) | Unknown | 4.5 |
| Sediment supply from the river, $q_0$ (m$^3$/y) | 1700 [45] | 1100 |
| Formation time, $t_0$ (yr) | 33 | 33 |
| Total depth of closure and berm height, $D_C + D_B$ (m) | 1.36 [45] | 1.36 |
| Length of the shoreline, $L$ (m) | 450 | 450 |

Figure 17 shows the formation of the shoreline at the boundary, which is the pier located at 490 m from the Funtasu River mouth, in which the red line represents the calculated shoreline evolution with the effect of the boundary, Equation (13), and the blue line shows the shoreline evolution without the effect of the boundary, Equation (6). It can be seen that the calculated result using Equation (13) shows better agreement with the measured data than the result of Equation (6). This indicates that the boundary has already affected the evolution of the left shoreline in the Funatsu River delta. In Figure 17, the measured data were obtained from an aerial photograph in 1982 and Google Earth images from 2006 to 2015. It should be noted that the data obtained from Google Earth from 2006 to 2015 are not continuous.

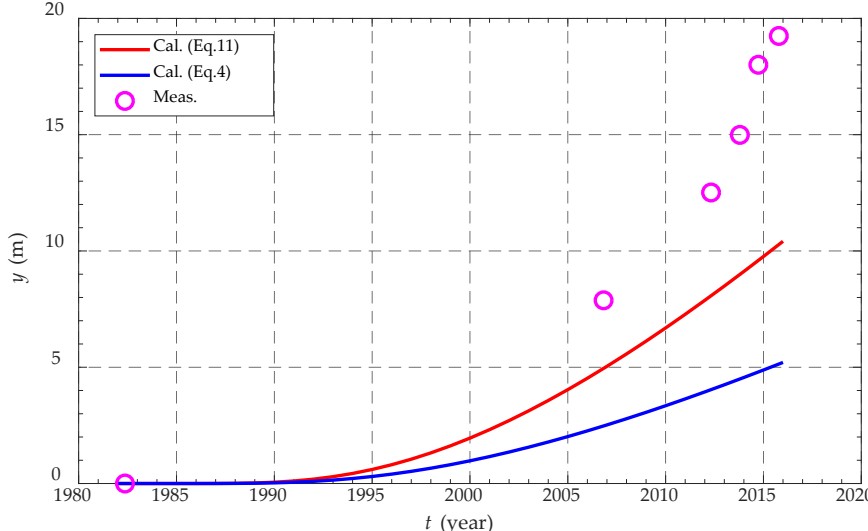

**Figure 17.** Formation of shoreline position at the boundary (the pier) in Funatsu River.

## 5. Classification of Delta with Finite- and Infinite-Beach Length

In the previous discussion for river-delta formation with analytical solutions for infinite and finite shorelines, two demarcations for the boundary to take effect were figured out as $t^* = 0.1$ at the boundary and $t^* = 0.3$ at the river mouth. It would, therefore, be useful to classify the river delta based on a demarcation from which suggestions for using an appropriate analytical solution can be given.

The dependence of $t^*$ on the temporal ($t$) and spatial ($L$) scales of the delta can be seen in Equation (17) as:

$$t = t^* \frac{L^2}{\varepsilon} \tag{33}$$

From the two demarcations of $t^* = 0.1$ at the boundary and $t^* = 0.3$ at the river mouth, two lines can be drawn with the slopes of 0.1 and 0.3. For a delta with given parameters of $t$, $L$, and $\varepsilon$, the values of $t$ and $L^2/\varepsilon$ can be plotted on the same diagram, with the two lines representing the demarcations when the boundary affects the shoreline evolution. Based on the location of the plotting point, with respect to the demarcation lines, it can be concluded that whether the boundary effect occurs in this delta or not.

Table 5 shows the spatial and temporal scales of the Ombrone River and the Funatsu River deltas, while Table 6 shows the spatial and temporal scales of the delta in the experiment of [2]. It should be noted that the time intervals are 200 years, 5 years, and 10 min for the Ombrone River, the Funatsu River, and the experiment, respectively.

**Table 5.** Spatial and temporal scales of the Ombrone River and the Funatsu River deltas.

| Ombrone River Delta | |
|---|---|
| $L = 6000$ m, $\varepsilon = 45$ m$^2$/day | |
| No. in Figure 18 | $t$ (yr) |
| 1 | 100 |
| 2 | 300 |
| 3 | 500 |
| 4 | 700 |
| 5 | 900 |
| **Funatsu River delta** | |
| $L = 490$ m, $\varepsilon = 4.5$ m$^2$/day | |
| No. in Figure 18 | $t$ (yr) |
| 1 | 5 |
| 2 | 10 |
| 3 | 15 |
| 4 | 20 |
| 5 | 25 |
| 6 | 30 |
| 7 | 35 |

**Table 6.** Spatial and temporal scales of the delta in the experiment of [2].

| Experiment Series A [2] | |
|---|---|
| $L = 400$ cm, $\varepsilon = 15$ cm$^2$/s | |
| No. in Figure 18 | $t$ (s) |
| 1 | 600 |
| 2 | 1200 |
| 3 | 1800 |
| 4 | 2400 |
| 5 | 3000 |
| 6 | 3600 |
| 7 | 4200 |
| 8 | 4800 |

Figure 18 shows the log–log plot of the data in Tables 5 and 6 and the two demarcations at $t^* = 0.1$ and $t^* = 0.3$. In which, the blue and red lines represent two demarcations at $t^* = 0.1$ and $t^* = 0.3$, respectively. The experimental data, the Funatsu River data and the Ombrone River data are presented by the blue circles, the purple diamonds and the red squares, respectively. It is clear in Figure 18 that in the early stages of the formation process, the sediment has not been transported to the boundary and the diffusion length is smaller than the shoreline length. After a sufficient time, the sediment has reached the boundary, and the shorelines become finite, as in the cases of Experiment [2], the Funatsu River delta, and the Ombrone River delta. In the case of the Funatsu River delta, after 35 years or at present, the boundary has not affected the shoreline position at the river mouth. Therefore, the shoreline position at the river mouth of the Funatsu River delta still follows the trend of the analytical solution for infinite shorelines.

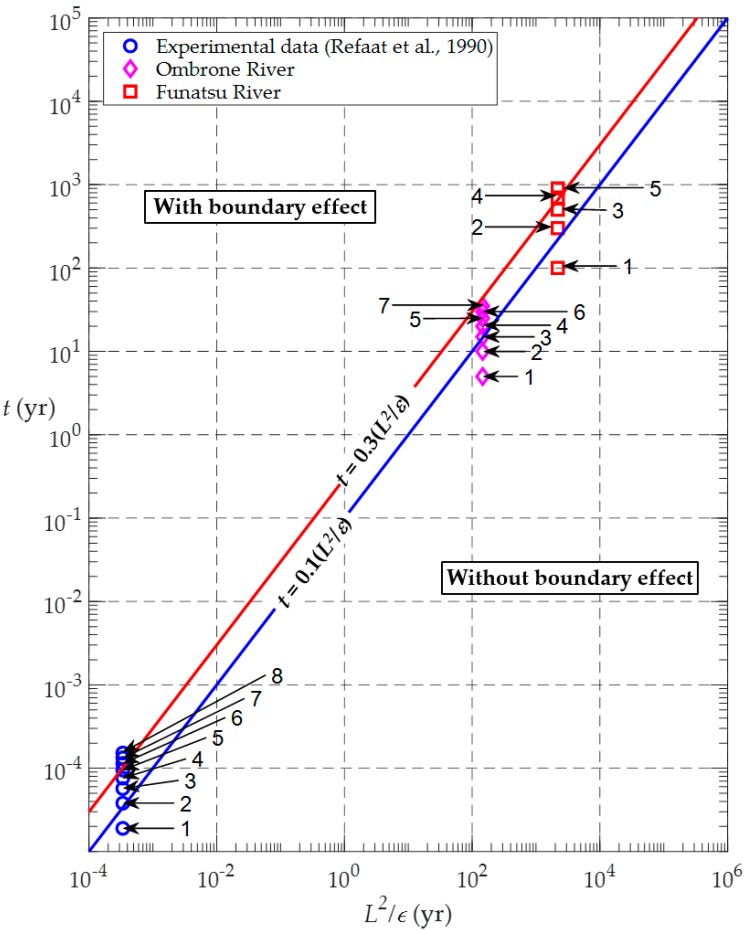

**Figure 18.** Demarcations for formation processes of river-delta shorelines.

## 6. Conclusions

An analytical solution for the formation of a river delta with finite shorelines has been developed and validated with experimental data and two river deltas with different scales in the world. Two demarcations are proposed, at $t^* = 0.1$ and $0.3$, for the classification of river-delta shorelines in two categories of finite and infinite shorelines. For the formation of river-delta shorelines in this study, in the early stages, the shorelines can be considered as infinite shorelines, so an analytical solution of [3] can be applied. However, after a sufficient time, the sediment from the river mouth has reached the boundaries, and the shoreline position should be modeled using the analytical solution developed in this study for the finite shorelines.

The new analytical solution proposed in this study can be used as a simple and economical means to take a quick, quantitative evaluation of delta-shoreline response

under the effects of lateral-boundary conditions such as groins or headlands. Although the closed-form solution presented in this study has been well-verified with one set of experimental data and two river deltas in Japan and Italy, there are still limitations in this approach. First, the longshore-sand-transport rate is assumed to occur uniformly. However, the longshore-sand-transport rate is proportional to the breaking-wave height along the coast. Hence, it is useful to make an improvement for the solution for non-uniform longshore-sediment transport. Second, the sediment supply from the river is considered to be constant and equal to $q_0$, but the sediment supply from the river always changes. Therefore, it is worth making an investigation into the case of non-uniform sediment supply from the river.

**Author Contributions:** Conceptualization, D.V.D., H.T., M.L. and N.T.V.; methodology, D.V.D. and H.T.; formal analysis, D.V.D. and H.T.; investigation, D.V.D. and H.T.; resources, D.V.D. and H.T.; data curation, D.V.D. and H.T.; writing—original draft preparation, D.V.D.; writing—review and editing, D.V.D., H.T., M.L. and N.T.V.; visualization, D.V.D., H.T., M.L. and N.T.V.; supervision, H.T., M.L. and N.T.V.; project administration, H.T., M.L. and N.T.V.; funding acquisition, H.T., M.L. and N.T.V. All authors have read and agreed to the published version of the manuscript.

**Funding:** M.L. wishes to express their grateful thanks to the Invitation Fellowship of the Japan Society for the Promotion of Science. The authors wish to express their grateful thanks for the contribution made by the members of the "Development of model systems to assess and forecast morphological changes and countermeasures to stabilize the beaches in the Mid-Central Vietnam region" project, funded by the Ministry of Science and Technology (MOST), Vietnam as the Decision No: 900/QĐ-BKHCN.

**Data Availability Statement:** Not applicable.

**Acknowledgments:** The authors wish to express their grateful thanks for the contribution made by the members of the "Development of model systems to assess and forecast morphological changes and countermeasures to stabilize the beaches in the Mid-Central Vietnam region" project, funded by the Ministry of Science and Technology (MOST), Vietnam.

**Conflicts of Interest:** The authors declare no conflict of interest.

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
