# Peer review of "A Theory for Estuarine Delta Formation with Finite Beach Length under Sediment Supplied from the River"

_jmse, doi:10.3390/jmse10070947_

Round 1
Reviewer 1 Report
The authors have presented analytical solutions for shoreline changes in estuaries. These described the formation of the finite river delta shoreline well and classified the formation processes. This paper is interesting; however, there are some issues. For this reason, I advise acceptance after major revision.
Major issues
- Chap. 3: Please add more detail about the experiment condition, such as D50, water discharge rate, and measure method for the topography. Additionally, the role of this chapter in the overall structure of the manuscript is unclear.
- 4.2: How did the substrate on the left shore of the Ombrane River (shown in Fig 13) affect shoreline progradation in the early stage? Did the present calculation express the impact?
- L288, L303, and L331: I could not find the fitting method and process in this manuscript. Which part does provide the information?
- L291: Which values were calibrated in Tab 2, which shows parameters from previous research?
- Tab 3: The audience might confuse about this list and the caption because some values are restated from Tab 2. For more readable information, the authors need to arrange these tables.
- L308-311: Are these equations only applicable to the Ombrone River's mechanisms? If this approach applies to general river mouths, this approach should be placed in Chap 2.
- 4.3: While this application approach is the same as 4.2, there are almost the same issues I indicated above.
- 4.2-4.4: Sections 4.2 and 4.3 describe agreements between theoretical calculation and measurements, and the manuscript does not have a discussion chapter. In addition, the result of Chap 3 appeared and was analyzed in 4.4. To improve the paper and be easy to understand explanation for the audience, I suggest restructuring these chapters and sections and adding a discussion.
Minor issues
- Some indent seems wrong, especially after equation lines (e.g., L82, L105, L111, L113, L139, and L142 until page 5). Please review the format.
Reviewer 2 Report
Thank you very much for this interesting paper. I have only few remarks:
22: ";" instead of ","
222: Are sand properties relevant for the results?
224: epsilon is e.g. depending on the wave hight. How did you choose this in the model?
Table1, Fig11: 6 or 6.4km
257: slopes instead of lopes?
Fig14: Please add legend for different colors
299: Did you apply the equations also for the right shoreline?
360: Link for Tab6 is missing
Reviewer 3 Report
Paper very well written! Fiure 18 needs a to be eventually retouched.
Reviewer 4 Report
Please see attached file.

Round 2
Reviewer 1 Report
Thank you for responding to the reviewer's comments with care. The authors added supporting words and restructured the manuscript. The reviewer thinks this improved paper is suitable for publication after the minor corrections regarding the paper format.
- The revised manuscript has no space between the figure captions and the body line.
- I guess the blue-colored words indicated the link; however, some linked characters seem wrong.